# Hepatic lipid overload triggers biliary epithelial cell activation via E2Fs

Ece Yildiz[1], Gaby El Alam[2†], Alessia Perino[1†], Antoine Jalil[1], Pierre-Damien Denechaud[3‡], Katharina Huber[3], Lluis Fajas[3,4], Johan Auwerx[2], Giovanni Sorrentino[1§], Kristina Schoonjans[1*]

[1]Laboratory of Metabolic Signaling, Institute of Bioengineering, Ecole Polytechnique Fédérale de Lausanne, Lausanne, Switzerland; [2]Laboratory of Integrative Systems Physiology, Institute of Bioengineering, Ecole Polytechnique Fédérale de Lausanne, Lausanne, Switzerland; [3]Center for Integrative Genomics, Université de Lausanne, Lausanne, Switzerland; [4]INSERM, Occitanie, Montpellier, France

**\*For correspondence:**
kristina.schoonjans@epfl.ch

†These authors contributed equally to this work

**Present address:** ‡Institute of Metabolic and Cardiovascular Diseases (I2MC), UMR1297, INSERM, University of Toulouse, Toulouse, France; §Department of Medical, Surgical and Health Sciences, University of Trieste, Trieste, Italy

**Competing interest:** The authors declare that no competing interests exist.

**Abstract** During severe or chronic hepatic injury, biliary epithelial cells (BECs) undergo rapid activation into proliferating progenitors, a crucial step required to establish a regenerative process known as ductular reaction (DR). While DR is a hallmark of chronic liver diseases, including advanced stages of non-alcoholic fatty liver disease (NAFLD), the early events underlying BEC activation are largely unknown. Here, we demonstrate that BECs readily accumulate lipids during high-fat diet feeding in mice and upon fatty acid treatment in BEC-derived organoids. Lipid overload induces metabolic rewiring to support the conversion of adult cholangiocytes into reactive BECs. Mechanistically, we found that lipid overload activates the E2F transcription factors in BECs, which drive cell cycle progression while promoting glycolytic metabolism. These findings demonstrate that fat overload is sufficient to reprogram BECs into progenitor cells in the early stages of NAFLD and provide new insights into the mechanistic basis of this process, revealing unexpected connections between lipid metabolism, stemness, and regeneration.

## Editor's evaluation

This important study reports that a high-fat diet induces biliary epithelial cell proliferation and suggests this may account for the so-called ductular reaction in advanced fatty liver disease. Convincing data support the finding that E2f1 is required for BEC proliferation in mice fed with HFD, and organoid models indicate that lipid abundance promotes glycolysis in an E2F-dependent manner.

## Introduction

Under physiological conditions, the hepatic epithelium, composed of hepatocytes and BECs (or cholangiocytes), is non-proliferative. Yet upon injury, these two cell types are capable of rapidly changing their phenotype from quiescent to proliferative, contributing to the prompt restoration of damaged tissue (**Gadd et al., 2020**; **Michalopoulos, 2014**; **Miyajima et al., 2014**; **Yanger and Stanger, 2011**). However, in chronic liver injury, characterized by impaired hepatocyte replication, BECs weigh-in and serve as the cell source for regenerative cellular expansion through the DR process (**Choi et al., 2014**; **Deng et al., 2018**; **Español-Suñer et al., 2012**; **Huch et al., 2013**; **Lu et al., 2015**; **Raven et al., 2017**; **Rodrigo-Torres et al., 2014**; **Russell et al., 2019**).

The molecular basis by which BECs expand during the DR has been extensively studied in models of chemical biliary damage and portal fibrosis using the chemical 3,5-diethoxycarbonyl-1,4-dihydroc

ollidine (DDC). Several signaling pathways involving YAP (*Meyer et al., 2020*; *Pepe-Mooney et al., 2019*; *Planas-Paz et al., 2019*), mTORC1 (*Planas-Paz et al., 2019*), TET1-mediated hydroxymethylation (*Aloia et al., 2019*) and NCAM1 (*Tsuchiya et al., 2014*) have been reported to drive this process. Importantly, DR has also been observed in late-stage NAFLD patients with fibrosis and portal inflammation (*Gadd et al., 2014*; *Sato et al., 2019*; *Sorrentino et al., 2005*). NAFLD, one of the most common chronic diseases, initiates with increased lipid accumulation, a stage called steatosis (*Paschos and Paletas, 2009*). This pathology progresses into inflammation and fibrosis that can cause cirrhosis and hepatocellular carcinoma, which are the most frequent liver transplantation indications (*Byrne and Targher, 2015*). YAP has been found to be activated in BECs in fibrotic livers but not in steatosis (*Machado et al., 2015*), suggesting that YAP activation is necessary to support DR in the late fibrotic NAFLD stages, and thus, leaving the early molecular mechanisms of BEC activation, which precede the onset of the DR, unexplored.

BEC-derived organoids (BEC-organoids) can be established from intrahepatic bile duct progenitors and can proliferate, representing a promising in vitro approach to study regenerative mechanisms and therapies (*Huch et al., 2015*; *Huch et al., 2013*; *Li et al., 2017*; *Okabe et al., 2009*; *Shin et al., 2011*; *Sorrentino et al., 2020*). These self-renewing bi-potent organoids express biliary progenitor markers and are capable of differentiating into functional cholangiocyte- and hepatocyte-like lineages in specific differentiation media, which can engraft and repair bile ducts (*Hallett et al., 2022*; *Sampaziotis et al., 2021*) and improve liver function when transplanted into a mouse with liver disease (*Huch et al., 2015*; *Huch et al., 2013*; *Li et al., 2017*).

Here, we used BEC-organoids and BECs isolated from chow diet (CD)- or high-fat diet (HFD)-fed mice and reported that they are affected by acute and chronic lipid overload, one of the initial steps of NAFLD. Lipid accumulation turns BECs from quiescent to proliferative cells, the earliest step of a DR, and promotes their expansion through the E2F transcription factors and the concomitant induction of glycolysis. These observations hence attribute a pivotal role to E2Fs, regulators of cell cycle and metabolism, in priming BEC activation during the early stages of NAFLD.

## Results

### BECs and BEC-organoids efficiently accumulate lipids in vivo and in vitro

To gain insight into how chronic lipid exposure, which induces liver steatosis, affects biliary progenitor function in vitro, we incubated single BECs with a mixture of oleic acid (OA) and palmitic acid (PA) (the fatty acid (FA) mix) – the two most abundant FAs found in livers of NAFLD patients (*Araya et al., 2004*), for 7 days and allowed BEC-organoid formation (*Figure 1A*). Surprisingly, we observed that BEC-organoids efficiently accumulated lipid droplets in a dose-dependent manner (*Figure 1B*), and this process did not affect organoid viability (*Figure 1C–D*). To investigate how cells adapt their metabolism to lipid overload, we monitored the expression of several genes involved in lipid metabolism, including *Scd1* (de novo lipogenesis) (*Figure 1E*), *Hmgcs2* (ketogenesis), *Pdk4* (inhibition of pyruvate oxidation), and *Aldh1a1* (prevention against lipid peroxidation products) (*Figure 1F*) and found it to be affected by FA addition. These results suggest that BEC organoids actively reprogram their metabolism to cope with aberrant lipid overload.

To determine whether the observed phenotype was preserved in fully formed organoids, we treated already established BEC-organoids with the FA mix for 4 days (*Figure 1—figure supplement 1A*). In line with our previous observations, BODIPY staining (*Figure 1—figure supplement 1B*), and triglyceride (TG) quantification (*Figure 1—figure supplement 1C*) showed a pronounced increase in lipid accumulation after 4 days, without affecting cell viability (*Figure 1—figure supplement 1D*).

To assess whether chronic lipid exposure affects BECs in vivo, we fed C57BL/6 J mice for 15 weeks with CD or HFD (*Figure 1G*) and analyzed their bile ducts. As expected, HFD-fed mice gained weight and developed liver steatosis, but no apparent fibrosis (*Figure 1—figure supplement 1E*). Of note, HFD-feeding led to an accumulation of lipid droplets in the periportal zone (*Figure 1—figure supplement 1F*) and within bile ducts, as reflected by the localization of BODIPY signal in PANCK (BEC marker) positive cells (*Figure 1H*), without inducing epithelial damage (*Figure 1—figure supplement 1G–I*). Moreover, flow cytometry analysis of BECs stained with EPCAM, a pan-BEC marker (*Aloia et al., 2019*; *Pepe-Mooney et al., 2019*; *Planas-Paz et al., 2019*) and BODIPY confirmed that these

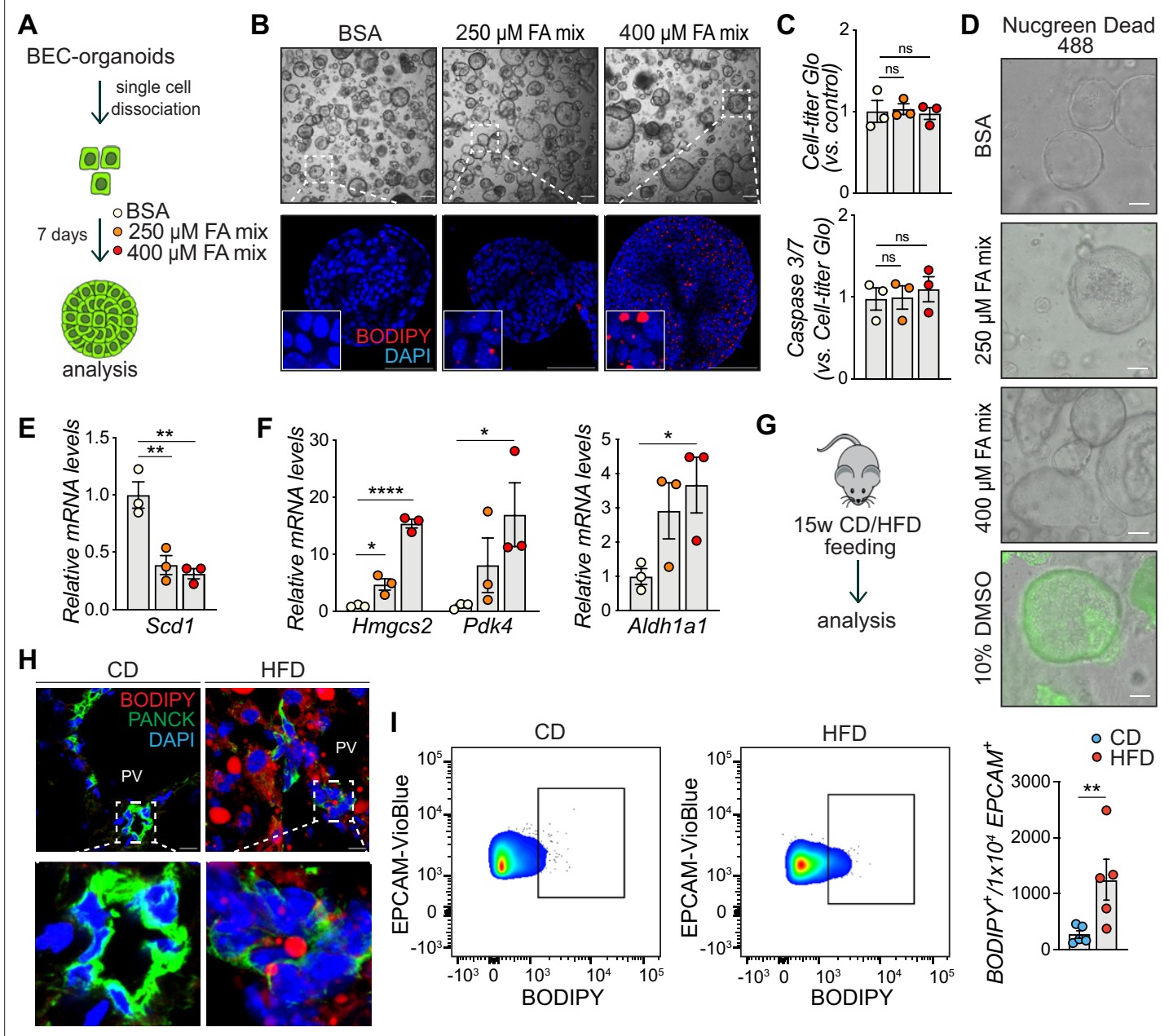

**Figure 1.** Biliary epithelial cells (BECs) accumulate lipids. (**A**) Schematic depicting fatty acid (FA) treatment of BEC-organoids in vitro. (**B**) Representative bright field and immunofluorescence (IF) images of lipids (BODIPY) in control (BSA) and FA-treated organoids. Close-up IF images were digitally zoomed in four times. n=3. (**C**) Cell-titer Glo and Caspase 3/7 activity measurement for viability and apoptosis detection relative to panel A. n=3. (**D**) Representative Nucgreen Dead 488 staining as composite images from bright field and fluorescent microscopy. n=3. (**E–F**) Quantification of *Scd1* (**E**) and *Hmgcs2*, *Pdk4*, and *Aldh1a1* (**F**) mRNA in control (BSA) and FA-treated organoids. n=3. (**G**) Schematic depicting chow diet (CD) and high-fat diet (HFD) feeding in vivo. (**H**) Representative images for co-staining of BODIPY and PANCK, relative to panel G. Close-up IF images were digitally zoomed in four times. n=3. (**I**) Representative quantitative plots of the percentage (left) and quantification (right) of BODIPY staining in EPCAM$^+$ BECs isolated from the liver of C57BL/6 J mice fed CD or HFD for 15 weeks. n=5. Data are shown as mean ± SEM. Absence of stars or ns, not significant (p>0.05); *p<0.05; **p<0.01; ****p<0.0001; one-way ANOVA with Dunnett's test (**C**), Fisher's LSD test (**E, F**), and unpaired, two-tailed Student's t-test (**I**) were used. PV, portal vein. Scale bars, 200 µm (**B** - bright field), 100 µm (**B** - IF, **D**), and 10 µm (**H**).

The online version of this article includes the following figure supplement(s) for figure 1:

**Figure supplement 1.** Further characterization of lipid accumulation in biliary epithelial cells (BECs).

cells are able to store lipids upon HFD feeding (*Figure 1I* and *Figure 1—figure supplement 1L*). Together, these in vitro and in vivo results demonstrate that BECs accumulate lipids upon chronic FA exposure, raising the question of the functional consequences of this previously unrecognized event on BEC behavior.

## HFD feeding promotes BEC activation and increases organoid formation capacity

To characterize in vivo the impact of chronic lipid overload on BECs at the molecular level, we isolated EPCAM⁺ BECs from livers of CD/HFD-fed mice by fluorescence-activated cell sorting (FACS) (*Figure 2A* and *Figure 2—figure supplement 1A*) and performed RNA sequencing (RNA-seq). Analysis of these data revealed a diet-dependent clustering in Principal Component Analysis (*Figure 2—figure supplement 1B*), indicating that HFD feeding induces considerable transcriptional changes in BECs in vivo. Differential expression analysis further revealed a total of 495 significantly changed genes, 121 upregulated and 374 downregulated (*Figure 2—figure supplement 1C* and *Supplementary file 1*). At the same time, HFD promoted the upregulation of *Ncam1* (*Figure 2—figure supplement 1D*), a well-established mediator of BEC activation. Consistent with the absence of fibrosis and biliary epithelial damage, canonical makers of the DR (*Manco et al., 2019*; *Sato et al., 2019*; *Tsuchiya et al., 2014*) were not changed in EPCAM⁺ BECs isolated from livers of HFD-fed mice (*Figure 2—figure supplement 1E*).

To further explore the transcriptional changes, we performed gene set enrichment analysis (GSEA) on Gene Ontology (GO) terms (*Figure 2B*) and KEGG (*Figure 2—figure supplement 1F*) pathways and identified cell proliferation, the most prevalent early feature of BEC activation (*Sato et al., 2019*) as the major process induced in these cells upon HFD feeding. Expansion of the reactive BECs requires detachment from their niche and invasion of the parenchyma toward the damaged hepatic area. This process is made possible by reorganizing the extracellular matrix (ECM) and reducing focal adhesion, effectively downregulated in EPCAM⁺ BECs upon HFD (*Figure 2B* and *Figure 2—figure supplement 1F*).

To validate the RNA-seq data, we monitored the activation of BECs in vivo by measuring the number of proliferating BECs in the portal region of the livers of mice fed either CD or HFD (*Figure 2C–D*). Of note, we found that HFD feeding was sufficient to induce a marked increase in the number of active BECs (i.e. Ki67⁺/OPN⁺ cells- *Figure 2C–D*). Similar results were observed in two independent cohorts of mice challenged with HFD and injected with EdU either to track proliferating cells by immunofluorescence (*Figure 2E–F*) or to quantify by flow cytometry the amount of EPCAM⁺/EdU⁺ BECs in the liver (*Figure 2G*), confirming that chronic lipid exposure stimulates the appearance of reactive BECs within the bile ducts.

The efficiency of BECs to generate organoids in vitro has been shown to mirror their activation status (*Aloia et al., 2019*). To functionally assess the impact of lipid overload on this process, we measured the organoid-forming capacity of isolated BECs, as a read-out of their regenerative potential. To this aim, we quantified the organoid formation efficiency of BECs isolated from CD- and HFD-fed mouse livers (*Figure 2H*). Strikingly, we observed that HFD-derived BECs were significantly more efficient in generating organoids than their CD counterparts (*Figure 2I*). Altogether, these results demonstrate that HFD feeding is sufficient to induce, in vivo, the exit of BECs from a quiescent state and the acquisition of both proliferative and pro-regenerative features.

## HFD feeding initiates BEC activation via E2Fs

To understand whether the mechanisms underlying BEC activation upon HFD in vivo involve canonical processes by which BECs expand during the DR in chronically damaged livers, we compared the transcriptional profile of BECs upon HFD with those of DDC-activated BECs (*Pepe-Mooney et al., 2019*; GSE125688). We identified the most pronounced changes shared between HFD and DDC samples by overlapping separate over-representation enrichment analyses (*Supplementary file 2*). Cell division, mitosis, and chromosome segregation were the shared enriched pathways for upregulated genes in HFD and DDC samples (*Figure 3A*), while ECM organization was the shared enriched pathway for the downregulated genes in HFD and DDC conditions (*Figure 3—figure supplement 1A*). We concluded that the mechanisms of BEC activation induced by lipid overload partially overlap with those by which BECs expand during the DR with biliary epithelial damage.

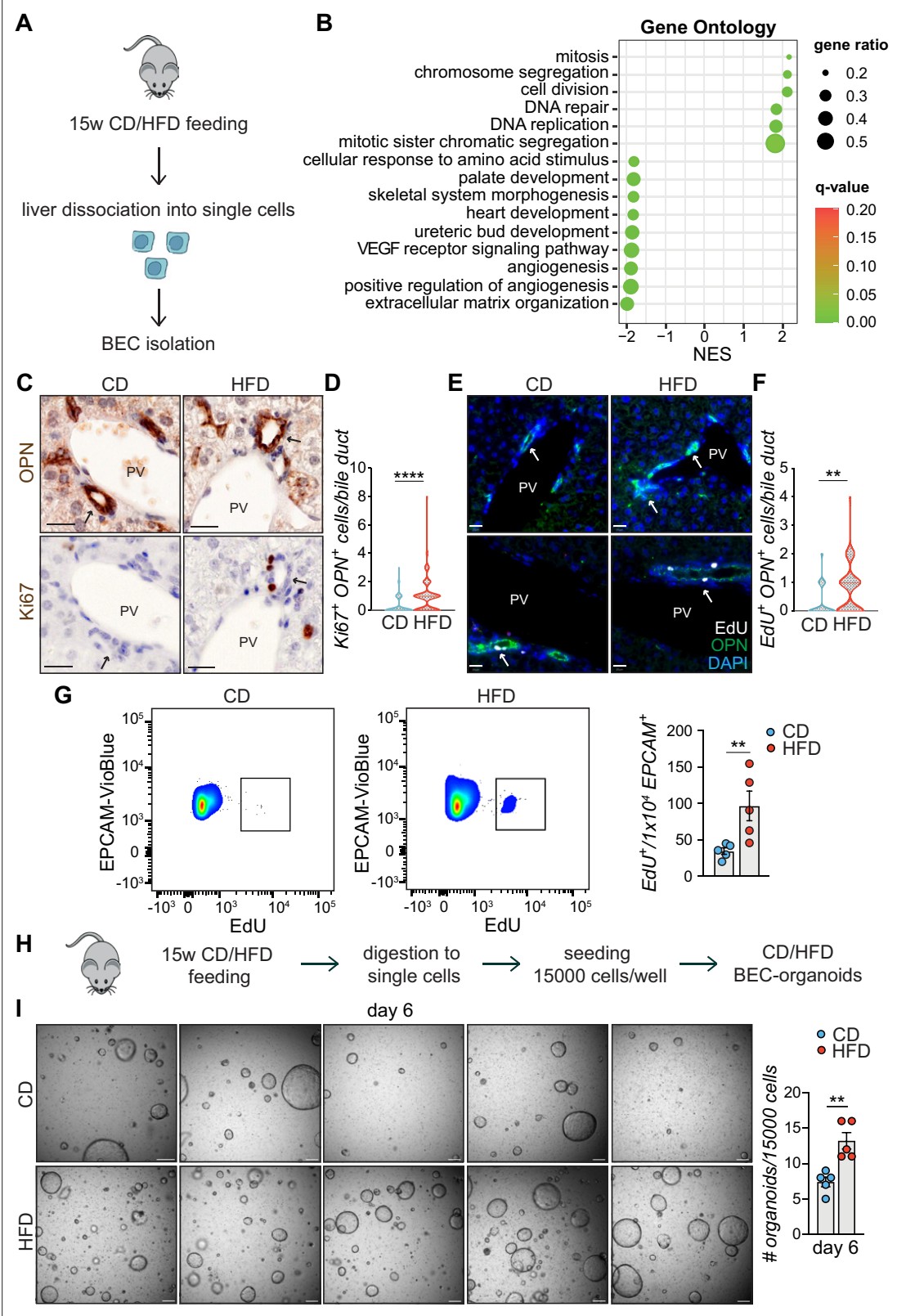

**Figure 2.** High-fat diet (HFD) feeding induces EPCAM+ biliary epithelial cell (BEC) proliferation. (**A**) Scheme depicting the isolation of EPCAM+ BECs from CD- and HFD-fed mice by fluorescence-activated cell sorting (FACS). (**B**) Gene set enrichment analysis (GSEA) of Gene Ontology (GO) terms. Top 15 upregulated biological processes (BP), ordered by normalized enrichment score (NES). q-value: false discovery rate adjusted p-values. (**C–F**) Representative co-staining images (**C, E**) and quantification (**D, F**) of BECs stained for OPN and Ki67 (**C–D**), or OPN and EdU (**E–F**) in livers of CD/

*Figure 2 continued on next page*

*Figure 2 continued*

HFD-fed mice. n=10 for Ki67 and n=5 for EdU. (**G**) Representative quantitative plots of the percentage (left) and quantification (right) of EdU⁺ EPCAM⁺ BECs, relative to panel A. n=5. (**H**) Schematic depicting BEC-organoid formation in vitro from CD/HFD-fed mouse livers. (**I**) Images of organoid colonies formed 6 days after seeding and quantification of organoids per well. n=5. Violin graphs depict the distribution of data points i.e the width of the shaded area represents the proportion of data located there. Other data are shown as mean ± SEM. **p<0.01; ****p<0.0001; unpaired, two-tailed Student's t-test was used. PV, portal vein. Arrowheads mark bile ducts. Scale bars, 20 μm (**C, E**), 200 μm (**I**).

The online version of this article includes the following figure supplement(s) for figure 2:

**Figure supplement 1.** RNA-seq analysis of EPCAM⁺ biliary epithelial cells (BECs) upon high-fat diet (HFD).

Of note, a more detailed analysis of DDC- and HFD-derived BECs, revealed the concomitant enrichment of four overlapping transcription factor (TF) gene sets, E2F1-4 (*Figure 3B*), and their target genes (*Figure 3—figure supplement 1B*), which have not been linked to BEC activation or DR previously. Moreover, we identified an enrichment of E2Fs (*Figure 3C* and *Supplementary file 2*) and cell division pathway (*Figure 3—figure supplement 1C*) as the most upregulated genes in proliferating BEC-organoids, further corroborating the role of E2Fs in these two in vitro (*Aloia et al., 2019*) and in vivo (*Pepe-Mooney et al., 2019*) DR models.

E2Fs are a large family of TFs with complex functions in cell cycle progression, DNA replication, repair, and G2/M checkpoints (*Dimova and Dyson, 2005*; *Dyson, 1998*; *Dyson, 2016*; *Ren et al., 2002*). Therefore, we hypothesized that the activation of E2Fs might represent an early event in the process of BEC activation, which is necessary for exiting the quiescent state and promoting BEC expansion. To test this hypothesis, we focused on E2F1, as it was the most enriched TF in our analysis, and assessed its role in BECs by feeding *E2f1⁺/⁺* and *E2f1⁻/⁻* mice with HFD (*Figure 3D*). Remarkably, *E2f1⁻/⁻* mice were refractory to BEC activation induced by lipid overload upon HFD, as opposed to *E2f1⁺/⁺* mice (*Figure 3E–F*). In addition, silencing of E2F1 in EPCAM⁺ BECs from livers of C57BL/6 J HFD-fed mice (*Figure 3—figure supplement 1D*) reduced the capacity to form organoids in vitro (*Figure 3—figure supplement 1E*). These results demonstrate a previously unrecognized role of E2F1 in controlling BEC activation during HFD-induced hepatic steatosis in vivo and support a pivotal role of this transcription factor in controlling BEC expansion.

## E2Fs promote BEC expansion by upregulating glycolysis

The exit of terminally-differentiated cells from their quiescent state requires both energy and building block availability to support cell proliferation. Proliferative cells, therefore, reprogram their glucose metabolism to meet their increased need for biomass and energy (*Vander Heiden et al., 2009*). Supporting this notion, our interrogation of in vitro BEC-organoid formation dataset (*Aloia et al., 2019*) revealed the enrichment of purine and pyrimidine metabolism, as well as the pentose-phosphate pathway, which is tightly connected to glycolysis (*Figure 4—figure supplement 1A*). In line with these findings, a substrate oxidation test in BEC-organoids revealed a preference for glucose, as reflected by the decrease in maximal respiration, when UK5099, a mitochondrial pyruvate carrier inhibitor, was used (*Figure 4A–B*), while no changes were observed with inhibitors of glutamine (BPTES) and FA (Etomoxir) metabolism (*Figure 4—figure supplement 1B–C*).

To investigate the metabolic changes in BEC-organoids upon HFD, we treated CD/HFD BEC-organoids with FA mix to mimic steatotic conditions in vitro (*Figure 4C*). We hypothesized that the presence of glucose and FA in culture media would reveal a metabolic shift of BEC-organoids. Consistent with our hypothesis, HFD-FA BEC-organoids demonstrated increased compensatory glycolytic rates (*Figure 4D–E* and *Figure 4—figure supplement 1D*). Of note, there was a reduction in oxidative phosphorylation in HFD-FA BEC-organoids, as evidenced by the decrease in maximal respiration (*Figure 4—figure supplement 1E–G*), which might reflect their preference for the glycolytic pathway to generate biomass.

Besides their prominent role in cell cycle progression, E2Fs coordinate several aspects of cellular metabolism (*Denechaud et al., 2017*; *Nicolay and Dyson, 2013*), and promote glycolysis in different contexts (*Blanchet et al., 2011*; *Denechaud et al., 2016*; *Huber et al., 2021*). These findings prompted us to postulate that E2F might control glycolysis and, thus, the glucose preference observed in BEC-organoids. To investigate this hypothesis, we treated BEC organoids with an E2F inhibitor, HLM006474 (*Figure 4F*). As expected, HLM006474 treatment reduced the transcriptional levels of several genes involved in cell cycle progression and glycolytic metabolism (*Figure 4G*) and decreased

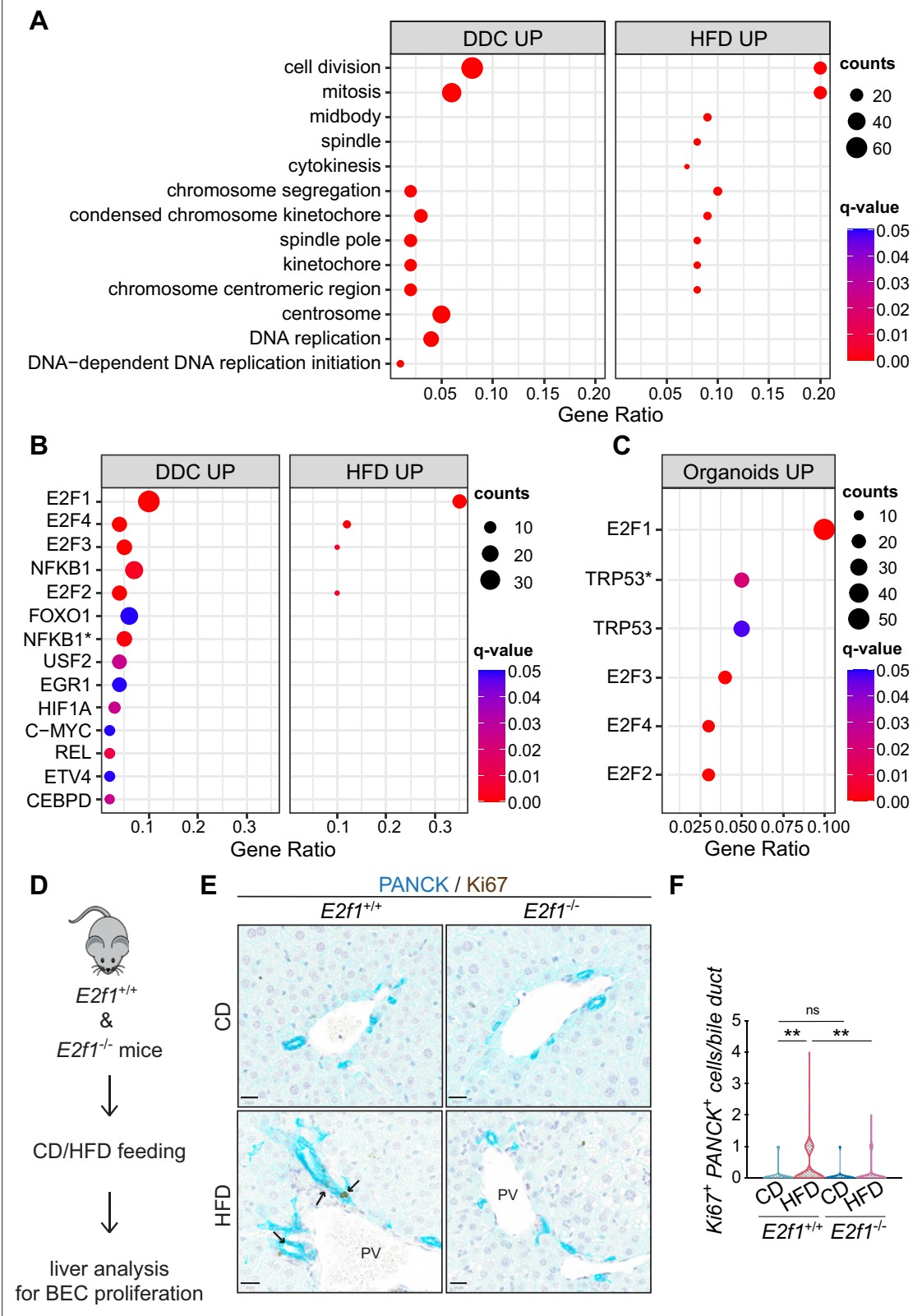

**Figure 3.** E2Fs are enriched in 3,5-diethoxycarbonyl-1,4-dihydrocollidine (DDC) and high-fat diet (HFD) datasets and mediate biliary epithelial cell (BEC) activation in vivo. (**A**) Over-representation analysis results. Top 13 enriched biological processes (BP) upon HFD (own data) and DDC (GSE125688) treatment. q-value: false discovery rate adjusted p-values, counts: number of found genes within a given gene set. (**B–C**) Enriched transcription factors (TFs) of upregulated genes identified by over-representation analysis in HFD (own data) and DDC (GSE125688) treatment (**B**), and during the process

*Figure 3 continued on next page*

*Figure 3 continued*

of organoid formation from single BECs (Organoids vs. T0) (GSE123133) (**C**). Asterisk (*) marks TFs of the 'TF_ZHAO' gene set. (**D**) Schematic depicting in vivo E2F1 analysis. (**E–F**) Representative images of PANCK/Ki67 co-staining in livers of *E2f1$^{+/+}$* and *E2f1$^{-/-}$* mice fed with chow diet (CD) or HFD (**E**) and quantification of proliferative BECs in the livers of the indicated mice (**F**). For CD, n=5 for *E2f1$^{+/+}$* and *E2f1$^{-/-}$*. For HFD, n=7 for *E2f1$^{+/+}$*, and n=8 for *E2f1$^{-/-}$*. Violin graphs depict the distribution of data points i.e., the width of the shaded area represents the proportion of data located there. ns, not significant; **p<0.01; two-way ANOVA with Tukey's test was used. PV, portal vein. Arrowheads mark bile ducts. Scale bars, 20 µm (**E**).

The online version of this article includes the following figure supplement(s) for figure 3:

**Figure supplement 1.** Extended analysis of biliary epithelial cells (BECs) upon high-fat diet (HFD), 3,5-diethoxycarbonyl-1,4-dihydrocollidine (DDC), during BEC-organoid formation and E2F1 silencing.

the glycolytic flux, as evidenced by the blunted proton efflux rate (PER) (*Figure 4H–I*). Moreover, E2F inhibition was able to reverse the metabolic phenotype only in HFD-FA BEC-organoids (*Figure 4J–K*).

In conclusion, these results demonstrate that HFD-induced E2F activation controls the conversion of BECs from quiescent to active progenitors by promoting the expression of cell cycle genes while simultaneously driving a shift toward glycolysis.

## Discussion

Through DR activation, BECs represent an essential reservoir of progenitors that are crucial for coordinating hepatic epithelial regeneration in the context of chronic liver diseases (*Choi et al., 2014*; *Deng et al., 2018*; *Español-Suñer et al., 2012*; *Huch et al., 2013*; *Lu et al., 2015*; *Raven et al., 2017*; *Rodrigo-Torres et al., 2014*; *Russell et al., 2019*). BEC functions are tightly controlled by YAP metabolic pathways (*Meyer et al., 2020*; *Pepe-Mooney et al., 2019*; *Planas-Paz et al., 2019*), and recent studies from different tissues have provided evidence that specific metabolic states play instructive roles in controlling cell fate and tissue regeneration (*Beyaz et al., 2016*; *Capolupo et al., 2022*; *Miao et al., 2020*; *Zhang et al., 2016*). Aberrant lipid accumulation is a hallmark of early NAFLD, and imbalances in lipid metabolism are known to affect hepatocyte homeostasis, including induction of lipo-toxicity and cell death (*Wang et al., 2016b*; *Sano et al., 2021*; *De Gottardi et al., 2007*; *Wobser et al., 2009*; *Ipsen et al., 2018*). However, the role of lipid dysregulation in BECs and whether it has an impact on BEC activation remains unexplored in the setting of NAFLD.

Here, using HFD- fed mouse models, we studied BEC metabolism in steatosis, the first stage of NAFLD, and demonstrated lipid accumulation in BECs during chronic HFD in vivo and their resistance to lipid-induced toxicity. By using BEC-organoids, we observed that FAs directly target BECs, without any involvement of hepatocytes and that BECs functionally respond to lipid overload. Importantly, BEC-organoids derived from CD- and HFD-fed mouse livers were shown to shift their cellular metabolism toward more glycolysis in the presence of lipids. Furthermore, we found that the HFD-induced metabolic shift was sufficient to reprogram BEC identity in vivo, allowing their exit from a quiescent state and the simultaneous acquisition of progenitor functions, such as proliferation and organoid-initiating capacity. These results highlight the metabolic plasticity of BECs and shed light on an unpredicted mechanism of BEC activation in HFD-induced hepatic steatosis. Importantly, we observed that lipid overload is sufficient to induce BEC activation in steatotic livers and that this process precedes parenchymal damage. While the functional contribution of lipid-activated BECs in liver regeneration during the late stages of NAFLD will require further studies, our data clearly point out the role of BECs as sensors and possibly effectors of early liver diseases such as steatosis.

To fully understand the underlying basis of the observed phenotype, we characterized the transcriptome of primary BECs derived from steatotic livers of HFD-fed mice and demonstrated that long-term feeding of a lipid-enriched diet strongly promotes BEC proliferation, suggesting a strong link between metabolic adaptation and progenitor function. By combining our transcriptomic analysis with data mining of publicly available DR datasets (*Aloia et al., 2019*; *Pepe-Mooney et al., 2019*), we identified the E2F transcription factors as master regulators of BEC activation in the context of NAFLD. Moreover, the expression of *Pdk4,* an E2F1 target (*Hsieh et al., 2008*; *Wang et al., 2016a*), was upregulated in FA-treated BEC-organoids and upon HFD. PDK4 limits the utilization of pyruvate for oxidative metabolism while enhancing glycolysis, which reinforces our data demonstrating that E2Fs rewire BEC metabolism toward glycolysis to fuel progenitor proliferation. These observations feature E2Fs as the molecular rheostat integrating the metabolic cell state with the cell cycle

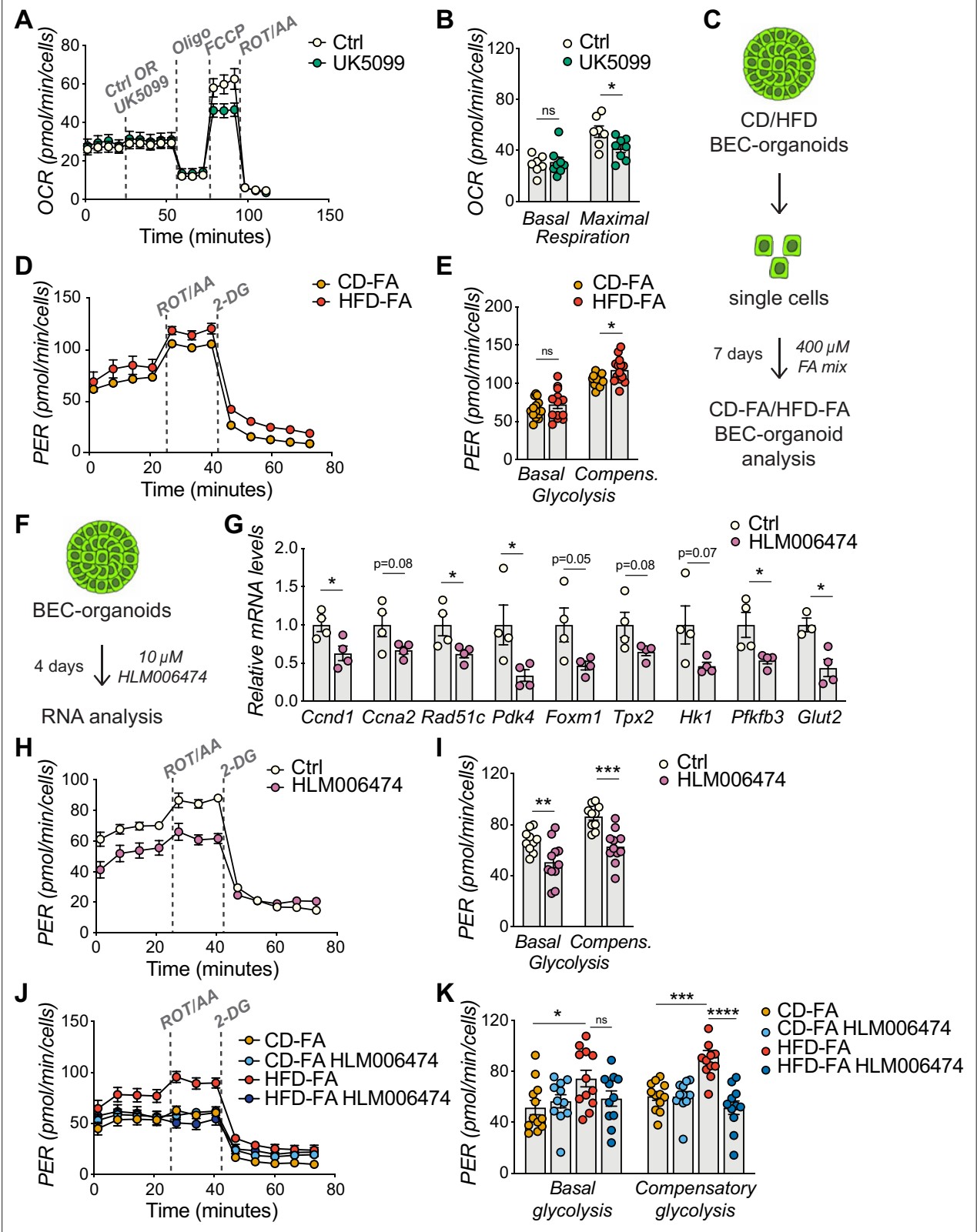

**Figure 4.** E2Fs promote glycolysis in biliary epithelial cell (BEC)-organoids. (**A–B**) Seahorse Substrate Oxidation Assay using UK5099, a mitochondrial pyruvate carrier inhibitor (**A**), and assessment of the glucose dependency (**B**) in CD-derived BEC-organoids. n=7 for control (Ctrl), n=8 for UK5099. (**C**) Scheme depicting the treatment of chow diet (CD)/high-fat diet (HFD)-derived BEC-organoids with fatty acid (FA) mix. (**D–E**) Proton efflux rate (PER) (**D**), and basal and compensatory (Compens.) glycolysis (**E**) were measured using Seahorse XF Glycolytic Rate Assay. Relative to panel C. n=14. (**F**) Scheme

*Figure 4 continued on next page*

*Figure 4 continued*

depicting the treatment of CD-derived BEC-organoids with E2F inhibitor, HLM006474. (**G**) RT-qPCR of selected cell cycle and glycolytic genes, relative to panel F. n=4. (**H–I**) PER during the Seahorse XF Glycolytic Rate Assay (**H**), and basal and compensatory (Compens.) glycolysis (**I**), relative to panel F. n=10 for control (Ctrl), n=11 for HLM006474. (**J–K**) PER during the Seahorse XF Glycolytic Rate Assay (**J**), and basal and compensatory glycolysis (**K**), relative to panel C and treatment with HLM006474. n=12 for CD-FA and HFD-FA, n=11 for HLM006474. Data are shown as mean ± SEM. Absence of stars or ns, not significant (p>0.05); *p<0.05; **p<0.01; ***p<0.001; ****p<0.0001; unpaired, two-tailed Student's t-test (**G**), and two-way ANOVA with Sidak's test (**B, E, I, K**) were used.

The online version of this article includes the following figure supplement(s) for figure 4:

**Figure supplement 1.** E2F activation correlates with increased glycolysis.

machinery to coordinate BEC activation. However, how E2Fs are regulated upon HFD and whether they are interconnected with the already known YAP, mTORC1, and TET1 pathways remain unknown and will require further investigation.

While our data are derived from obese mice, a recent report showed increased numbers of cholangiocytes in steatotic human livers (*Hallett et al., 2022*), and E2F1 has been found to be upregulated in the livers of obese patients (*Denechaud et al., 2016*). Moreover, human subjects with elevated visceral fat demonstrated increased glucose metabolism (*Broadfield et al., 2021*). These observations, while correlative, set the ground for future research in understanding the role and the therapeutic potential of lipid metabolism and E2Fs in controlling BEC activation and, thus, hepatic regeneration in humans.

## Materials and methods
### Mouse studies and ethical approval
All the animal experiments were authorized by the Veterinary Office of the Canton of Vaud, Switzerland, under license authorizations VD3721 and VD2627.b. C57BL/6JRj (in the text referred to as C57BL/6 J) mice were obtained from Janvier Labs and *E2f1*$^{+/+}$ and *E2f1*$^{-/-}$ (B6;129S4-E2f1tm1 Meg/J) mice were purchased from The Jackson Laboratory. 8-week-old C57BL/6 J male mice were fed with Chow Diet (CD - SAFE Diets, SAFE 150) or High Fat Diet (HFD - Research Diets Inc, D12492i) for 15 weeks. 7-week-old *E2f1*$^{+/+}$ and *E2f1*$^{-/-}$ male mice were fed with Chow Diet (CD – Kliba Nafag 3336) or High Fat Diet (HFD - Envigo, TD93075) for 29 weeks. The well-being of the animals was monitored daily, and body weight was monitored once per week until the end of the experiment. All mice had unrestricted access to water and food, and liver tissues were harvested at the end of the experiment.

### Data reporting
Mice were randomized into different groups according to their genotype. A previous HFD experiment was used to calculate the sample size for C57BL/6 J mouse experiments. Mice showing any sign of severity, predefined by the Veterinary Office of the Canton of Vaud, Switzerland, were sacrificed and excluded from the data analyses. In vitro experiments were repeated with at least three biological replicates (BEC-organoids from different mice) or were repeated at least twice by pooling four mice per condition (for Seahorse analysis).

### Proliferation assay
Cell proliferation was assessed by EdU assay (Click-iT EdU Alexa Fluor 647, ThermoFisher, C10340) following the manufacturer's instructions. For in vivo studies, EdU was resuspended in phosphate-buffered saline (PBS- ThermoFisher, 10010002), and 200 µl of the solution was injected intraperitoneally (50 µg per g of mouse weight) 16 hr before the sacrifice.

### EPCAM$^+$ BEC isolation, FACS, and flow cytometry analysis
23-week-old CD/HFD-fed C57BL/6 J male mice were used for this experiment and sacrificed in the fed state. To isolate the BECs, mouse livers were harvested and digested enzymatically, as previously reported (*Broutier et al., 2016*). Briefly, livers were minced and incubated at 37 °C for up to 2 hr in a digestion solution (1% fetal bovine serum [FBS; Merck/Sigma, F7524] in DMEM/Glutamax [ThermoFisher, 31966–021] supplemented with HEPES [ThermoFisher, 15630–056] and Penicillin/Streptomycin [ThermoFisher, 15140–122] containing 0.0125% [mg/ml] collagenase [Merck/Sigma,

C9407], 0.0125% [mg/ml] dispase II [ThermoFisher, 17105–041] and 0.1 mg/ml of DNAase [Merck/Sigma, DN25]). Digested livers were then dissociated into single cells with TrypLE [GIBCO, 12605028] and washed with washing buffer (1% FBS [Merck/Sigma, F7524] in Advanced DMEM/F-12 [GIBCO, 12634010] supplemented with Glutamax [ThermoFisher, 35050061], HEPES [ThermoFisher, 15630–056] and Penicillin/Streptomycin [ThermoFisher, 15140–122]).

For FACS analysis, single cells were filtered with a 40 µm cell strainer (Falcon, 352340) and incubated with fluorophore-conjugated antibodies CD45–PE/Cy7 (BD Biosciences, 552848), CD11b–PE/Cy7 (BD Biosciences, 552850), CD31–PE/Cy7 (Abcam, ab46733), CD31-PE/Cy7 (BD Biosciences, 561410) and EPCAM–APC (eBioscience, 17-5791-82) for 30 min on ice. BECs were sorted using FACSAria Fusion (BD Biosciences) as previously described (*Aloia et al., 2019*). Briefly, individual cells were sequentially gated based on cell size (forward scatter (FSC) versus side scatter (SSC)) and singlets. BECs were then selected based on EPCAM positivity after excluding leukocytes (CD45+), myeloid cells (CD11b+), and endothelial cells (CD31+), yielding a population of single CD45-/CD11b-/ CD31-/EPCAM+ cells.

For flow cytometry analysis, livers were dissociated as described above, and single cells were stained for 20 min with BODIPY 558/568 (Invitrogen, D38D35) on ice, followed by incubation with fluorophore-conjugated antibodies CD45–PE/Cy7 (BD Biosciences, 552848), CD11b–PE/Cy7 (BD Biosciences, 552850), CD31–PE/Cy7 (BD Biosciences, 561410), EPCAM–VioBlue (Miltenyi Biotec, 130-123-871), and EdU Alexa Fluor 488 (ThermoFisher, C10425) for 30 min on ice. After washing with 1% BSA in PBS, the Click-IT EdU reaction (ThermoFisher, C10425) was performed according to the manufacturer's instructions. Finally, BEC suspension was resuspended in 1% BSA in PBS, and analyzed using BD LSRFortessa (BD Biosciences). All flow cytometry data were analyzed with FlowJo v10.8 software (BD Life Sciences).

## Silencing of E2F1 in EPCAM+ BECs and organoid formation

To silence E2F1, $2 \times 10^4$ EPCAM+ BECs isolated from the livers of HFD-fed mice were transfected with a pool of four ON-TARGETplus siRNAs for *E2f1* (Horizon, L-044993-00-0005) or with scrambled siRNAs (Horizon, D-001810-10-05), using TransIT-X2 (Mirus, MIR6000) according to the manufacturer's instructions. Briefly, the cells and the TransIT-X2 mix were centrifuged at 600 g for 45 min at 32 °C and then incubated for 4 hr at 37 °C. The cell suspension was harvested and seeded in Matrigel in the isolation medium. Growth of BEC-organoids was followed with a luminescent MT Cell Viability Assay (Promega, G9711).

## RNA preparation from EPCAM+ BECs and bulk RNA-seq data analysis

RNA was isolated from sorted BECs using the RNeasy micro kit (QIAGEN, 74104), and the amount and quality of RNA were measured with the Agilent Tapestation 4200 (Agilent Technologies, 5067–1511). As a result, RNA-seq of five CD and seven HFD samples was performed by BGI with the BGISEQ-500 platform. FastQC was used to verify the quality of the reads (*Andrews, 2010*). No low-quality reads were present, and no trimming was needed. Alignment was performed against the mouse genome (GRCm38) following the STAR (version 2.6.0 a) manual guidelines (*Dobin et al., 2013*). The obtained STAR gene counts for each alignment were analyzed for differentially expressed genes using the R package DESeq2 (version 1.34.0) (*Love et al., 2014*). A threshold of 1 log2 fold change and adjusted p-value smaller than 0.05 were considered when identifying the differentially expressed genes. A principal component analysis (PCA) (*Lê et al., 2008*) was used to explore the variability between the different samples.

## Gene set enrichment analysis (GSEA)

We used the clusterProfiler R package (*Yu et al., 2012*) to conduct GSEA analysis on various gene sets. Gene sets were retrieved from http://ge-lab.org/gskb/ for *M. musculus*. We ordered the differentially expressed gene list by log2 (Fold-changes) for the analysis with default parameters.

## Over-representation enrichment analysis

All significantly changing genes (adjusted p-value <0.05 and an absolute fold change >1) were split into 2 groups based on the direction of the fold change (genes significantly up- & down-regulated). An over-representation analysis using the clusterProfiler R package was performed on each of the two groups to identify biologically overrepresented terms.

### Figure generation with R

The R packages ggplot2 (*Wickham, 2016*) retrieved from https://ggplot2.tidyverse.org and ggpubr were used to generate figures.

## Culture of mouse liver BEC-organoids from single bile duct cells of digested livers

BEC-organoids were established from bile ducts of C57BL/6 J male mice as previously described (*Broutier et al., 2016*; *Sorrentino et al., 2020*). Thus, the liver was digested as detailed above (EPCAM+ BEC isolation), and bile ducts were isolated and were pelleted by centrifugation at 200 rpm for 5 min at 4 °C and washed with PBS twice. The digested liver solution, including single bile duct cells, was then dissociated into single cells with TrypLE (GIBCO, 12605028). Isolated cells were resuspended in Matrigel (Corning, 356231) and cast in 10 µl droplets in 48-well plates. When gels were formed, 250 µl of isolation medium (IM- Advanced DMEM/F-12- Gibco,12634010) supplemented with Glutamax (ThermoFisher, 35050061), HEPES (ThermoFisher, 15630–056), Penicillin/Streptomycin (ThermoFisher, 15140–122), 1 X B27 (Gibco, 17504044), 1 mM *N*-acetylcysteine (Sigma-Aldrich, A9165), 10 nM gastrin (Sigma-Aldrich, G9145), 50 ng/ml EGF (Peprotech, AF-100–15), 1 µg/ml Rspo1 (produced in-house), 100 ng/ml FGF10 (Peprotech, 100–26), 10 mM nicotinamide (Sigma-Aldrich, N0636), 50 ng/ml HGF (Peprotech, 100–39), Noggin (100 ng/ml produced in-house), 1 µg/ml Wnt3a (Peprotech, 315–20), and 10 µM Y-27632 (Sigma, Y0503) was added to each well. Plasmids for Rspo1 and Noggin production were a kind gift from Joerg Huelsken. After the first 4 days, IM was replaced with the expansion medium (EM), which was the IM without Noggin, Wnt3a, and Y-27632. For passaging, organoids were removed from Matrigel for a maximum of one week after seeding and dissociated into single cells using TrypLE Express (Gibco, 12604013). Single cells were then transferred to fresh Matrigel. Passaging was performed in a 1:3 split ratio.

For the FA treatment of BEC-organoids, palmitic acid (Sigma, P0500) and oleic acid (Sigma, O1008) were dissolved in 100% ethanol into 500 and 800 µM stock solutions, respectively, and kept at –20 °C. For each experiment, palmitic acid and oleic acid were conjugated to 1% fatty acid-free bovine serum albumin (BSA) (Sigma, A7030) in EM through 1:2000 dilution each (*Malhi et al., 2006*). The concentration of vehicle, ethanol, was 0.1% ethanol in final incubations, and 1% fatty acid-free BSA in EM was used as the control for FA treatment.

## Liver immunohistochemistry (IHC) and immunofluorescence (IF)

For paraffin histology, livers were washed in PBS (Gibco, 10010023), diced with a razor blade, and fixed overnight in 10% formalin (ThermoFisher, 9990244) while shaking at 4 °C. The next day fixed livers were washed twice with PBS, dehydrated in ascending ethanol steps, followed by xylene, and embedded in paraffin blocks. 4 µm thick sections were cut from paraffin blocks, dewaxed, rehydrated, and quenched with 3% $H_2O_2$ for 10 min to block the endogenous peroxidase activity (for IHC). Antigen retrieval was performed by incubating the sections in 10 mM citrate buffer (pH 6.0) for 20 min at 95 °C. After the sections were cooled to room temperature, they were washed and blocked with blocking buffer (1% BSA (Sigma, A7906) and 0.5% Triton X-100 (Sigma, X100) in PBS) for 1 hr at room temperature. The primary antibodies anti-Ki67 (ThermoFisher, MA5-14520), anti-PANCK (Novusbio, NBP600-579), anti-OPN (R&D Systems, AF808), anti-Cleaved caspase-3 (Cell Signaling, 9661) were diluted in a 1:100 dilution of the blocking buffer and incubated overnight at 4 °C. For IHC, ImmpRESS HRP conjugated secondary (VectorLabs MP-74-01-15 and MP-74-02-15) were incubated for 30 min, and detection was performed by using a 3.3'-diaminobenzidine (DAB) reaction. Sections were counterstained with Harris and mounted. For IF, sections were washed and incubated for 1 hr with Alexa Fluor conjugated secondary antibodies (1:1000 in blocking solution; Invitrogen). Following extensive washing, sections were counterstained with DAPI (ThermoFisher, 62248) and mounted in ProLong Gold Antifade Mountant (Thermo Fischer, P36930).

For IF of liver cryosections, the livers were frozen in O.C.T. compound (VWR chemicals) on dry ice filled with isopentane. 10 µm liver sections were cut from O.C.T embedded samples, hydrated, and washed twice in PBS. The sections were blocked in a blocking buffer for 1 hr at room temperature and incubated with BODIPY for 20 min. After fixation with 4% paraformaldehyde (PFA) solution (Sigma, 1004960700) for 15 min, sections were washed with PBS. Then, sections were permeabilized using 5% BSA in TBS-T and stained with primary antibody anti-PANCK diluted in blocking buffer for 16 hr at

4 °C. The next day, the sections were washed three times with PBS, and the appropriate Alexa Fluor secondary antibodies were diluted in blocking buffer (1:1000) and incubated with the sections for 1 hr at room temperature. The sections were washed in PBS and incubated with DAPI diluted 1:1000 in PBS for 1 hr at room temperature. Finally, the sections were mounted in ProLong Gold Antifade Mountant.

Stained sections were imaged by a virtual slide microscope (VS120, Olympus) and a confocal microscope (SP8, Leica). The image analysis was performed using QuPath (*Bankhead et al., 2017*) and Fiji software.

## BEC-organoid whole-mount immunofluorescence

BEC-organoids were incubated with BODIPY 558/568 for 20 min and then washed with PBS and extracted from Matrigel using Cell Recovery Solution (Corning, 354253). After fixing with 4% PFA in PBS (30 min, on ice), they were pelleted by gravity to remove the PFA and were washed with PBS and ultra-pure water. BEC-organoids were then spread on glass slides and allowed to attach by drying. The attached BEC-organoids were rehydrated with PBS and permeabilized with 0.5% Triton X-100 in PBS (1 hr, room temperature) and blocked for 1 hr in a blocking buffer. After washing with PBS, samples were incubated for 1 hr at room temperature with Alexa Fluor Phalloidin 488 (Invitrogen, A12379). Following extensive washing, samples were counterstained with DAPI and were imaged by a confocal microscope (LSM 710, Zeiss). Signal intensity was adjusted on each channel using Fiji software (*Schindelin et al., 2012*).

## Quantitative real-time qPCR for mRNA quantification

BEC-organoids were extracted from Matrigel using Cell Recovery Solution (Corning, 354253). RNA was extracted from organoid pellets using the RNAqueous total RNA isolation kit (Invitrogen, AM1931) and the RNeasy Micro Kit (Qiagen, 74004) following the manufacturer's instructions. RNA was transcribed to complementary DNA using QuantiTect Reverse Transcription Kit (Qiagen, 205314) following the manufacturer's instructions. PCR reactions were run on the LightCycler 480 System (Roche) using SYBR Green (Roche, 4887352001) chemistry. Real-time quantitative polymerase chain reaction (RT-qPCR) results were presented relative to the mean of *36b4* (comparative ΔCt method). Primers for RT-qPCR are listed in *Supplementary file 3*.

## E2F inhibition

For the E2F inhibition experiment, single BECs were grown for 7 days and allowed to form organoids. For the Seahorse experiment, BEC-organoids were treated with E2F inhibitor, HLM006474 (10 µM, Merck, 324461), overnight before the metabolic assay. For RT-qPCR analysis, BEC-organoids were treated with HLM006474 chronically for 4 days.

## Bioenergetics with Seahorse extracellular flux analyzer

The oxygen consumption rate (OCR), extracellular acidification rate (ECAR), and proton-efflux rate (PER) of the BEC-organoids were analyzed by an XFe96 extracellular flux analyzer (Agilent) following the manufacturer's instructions according to assay type.

For Mito Stress Test on CD/HFD-derived BEC-organoids, the organoids were grown with FA mix for 7 days. On day 7, 10 µM HLM006474 or DMSO as vehicle were added overnight. The next morning, BEC-organoids were dissociated, and 20,000 cells were seeded with Seahorse Assay Medium in XFe96 Cell Culture Microplates (Agilent, 101085–004), which were previously coated with 10% Matrigel in Advanced DMEM/F-12. Seahorse Assay Medium was unbuffered, serum-free pH 7.4 DMEM supplemented with 10 mM glucose (Agilent, 103577–100), 10 mM pyruvate (Gibco, 11360070), and 2 mM glutamine (Agilent, 103579–100), and 10 µM HLM006474 or DMSO (vehicle) were added when indicated. After 2 hr incubation for cell attachment, plates were transferred to a non-CO$^2$ incubator at 37 °C for 45 min. Mitochondrial OCR was measured in a time course before and after the injection of 1.5 µM Oligomycin (Millipore, 495455), 2.5 µM FCCP (Sigma, C2920), and 1 µM Rotenone (Sigma, R8875)/Antimycin A (Sigma, A8674).

For Glycolytic Rate Assay, CD BEC-organoids were grown without FA mix, and CD/HFD- derived BEC-organoids were grown with FA mix for 7 days. The Seahorse assay preparations, including the

E2F inhibitor were the same as mentioned above. GlycoPER was measured in a time course before and after the injection of 1 μM Rotenone/Antimycin A and 50 mM 2-DG (Sigma, D8375).

For the Substrate Oxidation Assay, CD BEC-organoids were grown without FA mix for 7 days. On day 8, they were dissociated and prepared for Seahorse assay without E2F inhibitor. Mitochondrial OCR was measured in a time course before and after the injection of Oligomycin (1.5 μM), FCCP (2.5 μM), and Rotenone/Antimycin A (1 μM) with or without UK5099 (Sigma, PZ0160), Etomoxir (Sigma, E1905) and BPTES (Sigma, SML0601), inhibitors of glucose oxidation, fatty acid oxidation and glutamine oxidation, respectively, in separate experiments.

All Seahorse experiments were normalized by cell number through injection of 10 μM of Hoechst (ThermoFisher, 62249) in the last Seahorse injection. Hoechst signal (361/486 nm) was quantified by SpectraMax iD3 microplate reader (Molecular Devices).

## BEC-organoid growth assay

BEC-organoid formation efficiency was quantified by counting the total number of cystic/single layer (lumen-containing) CD/HFD-derived BEC-organoids 6 days after seeding and normalizing it to the total number of cells seeded initially (15,000 cells). Organoids were imaged by DM IL LED inverted microscope (Leica), selected as regions of interest (ROI) using widefield 4 x magnification, and counted manually.

## BEC-organoid functional analysis

Grown BEC-organoids were treated with the FA mix for 4 days, and triglyceride levels were measured with a Triglyceride kit (Abcam, ab65336) following the manufacturer's instructions. Cell-titer Glo (Promega, G7570) was used to investigate cell viability. For functional assays involving single BECs, grown organoids were dissociated into single cells. 10,000 BECs were seeded, and organoid formation was allowed for 7 days. Cell viability, apoptosis, and cell death were investigated using Cell-titer Glo, Caspase 3/7 activity (Promega, G8091), and Nucgreen Dead 488 staining (Invitrogen, R37109), respectively, according to the protocol of manufacturers. For cell death staining, organoids were imaged using ECLIPSE Ts2 inverted microscope (Nikon).

## Quantification and statistical analysis

Data were presented as mean ± standard error of the mean (mean ± SEM). $n$ refers to biological replicates and is represented by the number of dots in the plot or stated in the figure legends. For the Seahorse experiments, $n$ refers to technical replicates pooled from 4 biological replicates and is represented by the number of dots in the plot or stated in the figure legends. The statistical analysis of the data from bench experiments was performed using Prism (Prism 9, GraphPad). The differences with $p < 0.05$ were considered statistically significant. No samples (except outliers) or animals were excluded from the analysis. Data are expected to have a normal distribution.

For two groups comparison, data significance was analyzed using a two-tailed, unpaired Student's t-test. In case of comparisons between more than two groups, one- or two-way ANOVA was used. Dunnet's, Tukey's, or Sidak's tests were used to correct for multiple comparisons. Statistical details of each experiment can be found in the respective figure legends.

# Acknowledgements

We thank Sabrina Bichet, Fabiana Fraga, and Jéromine Imbach for technical assistance, the EPFL-SV core facilities for support, and Yu Sun for critically reviewing the manuscript. This work was funded by the Ecole Polytechnique Fédérale de Lausanne (EPFL), the Kristian Gerhard Jebsen Foundation, the Swiss National Science Foundation (SNSF N° 310030_189178 to KS; SNSF N° 31003 A_179435 to JA), and AIRC Start-Up 2020 - ID.24322 to GS.

## Additional information

### Funding

| Funder | Grant reference number | Author |
|---|---|---|
| École Polytechnique Fédérale de Lausanne | | Kristina Schoonjans |
| Kristian Gerhard Jebsen Foundation | | Kristina Schoonjans |
| Swiss National Science Foundation | 310030_189178 | Kristina Schoonjans |
| Swiss National Science Foundation | 31003A_179435 | Johan Auwerx |
| AIRC Start-Up 2020 | 24322 | Giovanni Sorrentino |

The funders had no role in study design, data collection and interpretation, or the decision to submit the work for publication.

### Author contributions

Ece Yildiz, Conceptualization, Data curation, Formal analysis, Validation, Investigation, Visualization, Methodology, Writing – original draft, Writing – review and editing; Gaby El Alam, Data curation, Software, Formal analysis, Investigation, Visualization, Writing – review and editing; Alessia Perino, Supervision, Visualization, Project administration, Writing – review and editing; Antoine Jalil, Investigation, Visualization, Methodology, Writing – review and editing; Pierre-Damien Denechaud, Katharina Huber, Lluis Fajas, Johan Auwerx, Resources, Writing – review and editing; Giovanni Sorrentino, Conceptualization, Supervision, Writing – review and editing; Kristina Schoonjans, Conceptualization, Resources, Supervision, Funding acquisition, Investigation, Project administration, Writing – review and editing

### Author ORCIDs

Alessia Perino http://orcid.org/0000-0002-5434-3266
Pierre-Damien Denechaud http://orcid.org/0000-0003-3502-4814
Lluis Fajas http://orcid.org/0000-0002-1283-9503
Kristina Schoonjans http://orcid.org/0000-0003-1247-4265

### Ethics

All the animal experiments were authorized by the Veterinary Office of the Canton of Vaud, Switzerland, under license authorizations VD3721 and VD2627.b.

### Decision letter and Author response

Decision letter https://doi.org/10.7554/eLife.81926.sa1
Author response https://doi.org/10.7554/eLife.81926.sa2

## Additional files

### Supplementary files

• Supplementary file 1. Differential expression analysis results of EPCAM$^+$ BECs upon HFD. Related to *Figure 2*.

• Supplementary file 2. Over-representation analysis of upregulated and downregulated genes in HFD, DDC, and BEC-organoid formation datasets. Related to *Figure 3*.

• Supplementary file 3. Primers used for qPCR analysis.

• MDAR checklist

### Data availability

Computational analysis was performed using established packages mentioned in the Materials and methods, and no new code was generated. RNA-Seq data have been deposited in GEO under accession code GSE217739. Two publicly available RNA-Seq datasets of mouse BECs with accession

numbers GSE123133 (*Aloia et al., 2019*) and GSE125688 (*Pepe-Mooney et al., 2019*) were downloaded from the GEO and used for GSEA and over-representation enrichment analysis as mentioned previously. Source code is available at https://github.com/auwerxlab/Yildiz_eLife_2023 (copy archived at *Alam, 2023*).

The following dataset was generated:

| Author(s) | Year | Dataset title | Dataset URL | Database and Identifier |
| --- | --- | --- | --- | --- |
| Schoonjans K | 2023 | Hepatic lipid overload triggers biliary epithelial cell activation via E2Fs | http://www.ncbi.nlm.nih.gov/geo/query/acc.cgi?acc=GSE217739 | NCBI Gene Expression Omnibus, GSE217739 |

The following previously published datasets were used:

| Author(s) | Year | Dataset title | Dataset URL | Database and Identifier |
| --- | --- | --- | --- | --- |
| Aloia L, Vernaz G, Huch M | 2019 | Transcriptonal and epigenetic changes of adult liver cells in vivo and in vitro | https://www.ncbi.nlm.nih.gov/geo/query/acc.cgi?acc=GSE123133 | NCBI Gene Expression Omnibus, GSE123133 |
| Pepe-Mooney BJ, Dill MT, Alemany A, Ordovas-Montanes J, Matsushita Y, Rao A, Sen A, Miyazaki M, Anakk S, Dawson P, Ono N, Shalek AK, van Oudenaarden A, Camargo FD | 2019 | Single-Cell Analysis of the Liver Epithelium Reveals Dynamic Heterogeneity and an Essential Role for YAP in Homeostasis and Regeneration | https://www.ncbi.nlm.nih.gov/geo/query/acc.cgi?acc=GSE125688 | NCBI Gene Expression Omnibus, GSE125688 |

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

# Appendix 1

**Appendix 1—key resources table**

| Reagent type (species) or resource | Designation | Source or reference | Identifiers | Additional information |
| --- | --- | --- | --- | --- |
| Antibody | Anti-mouse CD45–PE/Cy7 (Rat monoclonal) | BD Biosciences | 552848 | FACS, Flow cytometry (1:100) |
| Antibody | Anti-mouse CD11b–PE/Cy7 (Rat monoclonal) | BD Biosciences | 552850 | FACS, Flow cytometry (1:100) |
| Antibody | Anti-mouse/human CD31–PE/Cy7 (Rat monoclonal) | Abcam | ab46733 | FACS (1:100) |
| Antibody | Anti-mouse CD31–PE/Cy7 (Rat monoclonal) | BD Biosciences | 561410 | FACS, Flow cytometry (1:100) |
| Antibody | Anti-mouse EPCAM–APC (Rat monoclonal) | eBioscience | 17-5791-82 | FACS (1:100) |
| Antibody | Anti-mouse EPCAM–VioBlue (Rat monoclonal) | Miltenyi Biotec | 130-123-871 | Flow cytometry (1:100) |
| Antibody | Anti-Ki67 (Rabbit monoclonal) | ThermoFisher | MA5-14520 | IF (1:100) |
| Antibody | Anti-PANCK (Rabbit polyclonal) | Novusbio | NBP600-579 | IF (1:100) |
| Antibody | Anti-OPN (Goat polyclonal) | R&D Systems | AF808 | IF (1:100) |
| Antibody | Anti-Cleaved caspase 3 (Rabbit polyclonal) | Cell Signaling | 9661 | IF (1:100) |
| Chemical compound, drug | Alexa Fluor Phalloidin 488 | Invitrogen | A12379 | IF (1:1000) |
| Chemical compound, drug | BODIPY 558/568 | Invitrogen | D38D35 | IF (5 µM) FACS (40 nM) |
| Chemical compound, drug | $N$-acetylcysteine | Sigma-Aldrich | A9165 | (1 mM) |
| Chemical compound, drug | Nicotinamide | Sigma-Aldrich | N0636 | (10 mM) |
| Chemical compound, drug | Y-27632 | Sigma-Aldrich | Y0503 | (10 µM) |
| Chemical compound, drug | HLM006474 | Merck | 324461 | (10 µM) |
| Chemical compound, drug | Oligomycin | Millipore | 495455 | (5 µM) |
| Chemical compound, drug | FCCP | Sigma-Aldrich | C2920 | (2.5 µM) |

*Appendix 1 Continued on next page*

*Appendix 1 Continued*

| Reagent type (species) or resource | Designation | Source or reference | Identifiers | Additional information |
|---|---|---|---|---|
| Chemical compound, drug | Rotenone | Sigma-Aldrich | R8875 | (1 µM) |
| Chemical compound, drug | Antimycin A | Sigma-Aldrich | A8674 | (1 µM) |
| Chemical compound, drug | 2-DG | Sigma-Aldrich | D8375 | (50 mM) |
| Chemical compound, drug | UK5099 | Sigma-Aldrich | PZ0160 | (2 µM) |
| Chemical compound, drug | Etomoxir | Sigma-Aldrich | E1905 | (4 µM) |
| Chemical compound, drug | BPTES | Sigma-Aldrich | SML0601 | (3 µM) |
| Commercial assay or kit | Anti-rabbit IgG Polymer Detection Kit (ImmPRESS Horse HRP conjugated secondary) | VectorLabs | MP-74-01-15 | IF (100 µl) per tissue section |
| Commercial assay or kit | Anti-mouse IgG Polymer Detection Kit (ImmPRESS Horse HRP conjugated secondary) | VectorLabs | MP-74-02-15 | IF (100 µl) per tissue section |
| Commercial assay or kit | Click-iT EdU Alexa Fluor 647 | ThermoFisher | Cat. # C10340 | (50 µg per g of mouse weight) |
| Commercial assay or kit | EdU Alexa Fluor 488 | ThermoFisher | C10425 | |
| Commercial assay or kit | MT Cell Viability Assay | Promega | G9711 | |
| Commercial assay or kit | RNeasy micro kit | QIAGEN | 74104 | |
| Commercial assay or kit | RNAqueous total RNA isolation kit | Invitrogen | AM1931 | |
| Commercial assay or kit | QuantiTect Reverse Transcription Kit | Qiagen | 205314 | |
| Commercial assay or kit | SYBR Green | Roche | 4887352001 | |
| Commercial assay or kit | Triglyceride kit | Abcam | ab65336 | |
| Commercial assay or kit | Cell-titer Glo | Promega | G7570 | |
| Commercial assay or kit | Caspase 3/7 activity | Promega | G8091 | |
| Commercial assay or kit | Nucgreen Dead 488 | Invitrogen | R37109 | |
| Peptide, recombinant protein | Gastrin | Sigma-Aldrich | G9145 | (10 nM) |

*Appendix 1 Continued on next page*

*Appendix 1 Continued*

| Reagent type (species) or resource | Designation | Source or reference | Identifiers | Additional information |
|---|---|---|---|---|
| Peptide, recombinant protein | EGF | Peprotech | AF-100–15 | (50 ng/ml) |
| Peptide, recombinant protein | FGF10 | Peprotech | 100–26 | (100 ng/ml) |
| Peptide, recombinant protein | HGF | Peprotech | 100–39 | (50 ng/ml) |
| Peptide, recombinant protein | Wnt3a | Peprotech | 315–20 | (1 μg/ml) |
| Strain, strain background (*Mus musculus*) | C57BL/6JRj | Janvier Labs | C57BL/6JRj | Males, 8-week-old |
| Strain, strain background (*Mus musculus*) | *E2f1+/+* and *E2f1-/-* | The Jackson Laboratory | (B6;129S4-E2f1tm1 Meg/J) | Males, 8-week-old |
| Software, algorithm | FlowJo | FlowJo | v10.8 | |
| Software, algorithm | Prism | GraphPad | Prism 9 | |
| Software, algorithm | STAR | *Dobin et al., 2013* | version 2.6.0 a | |
| Software, algorithm | DESeq2 | *Love et al., 2014* | version 1.34.0 | |
| Software, algorithm | clusterProfiler | *Yu et al., 2012* | | |
| Software, algorithm | ggplot2 | *Wickham, 2016* | | |
| Software, algorithm | QuPath | *Bankhead et al., 2017* | version 0.2.3 | |
| Software, algorithm | Fiji | *Schindelin et al., 2012* | version 2.3.0 | |
| Other | Chow Diet | SAFE | SAFE 150 | Section 'Mouse studies and ethical approval' For C57BL/6JRj |
| Other | High Fat Diet | Research Diets Inc | D12492i | Section 'Mouse studies and ethical approval' For C57BL/6JRj |
| Other | Chow Diet | Kliba Nafag | 3336 | Section 'Mouse studies and ethical approval' For *E2f1+/+* and *E2f1-/-* |
| Other | High Fat Diet | Envigo | TD93075 | Section 'Mouse studies and ethical approval' For *E2f1+/+* and *E2f1-/-* |
| Other | BEC-organoids from mouse | This paper | Original protocol: *Broutier et al., 2016* | Section 'Culture of mouse liver BEC-organoids from single bile duct cells of digested livers' |

*Appendix 1 Continued on next page*

*Appendix 1 Continued*

| Reagent type (species) or resource | Designation | Source or reference | Identifiers | Additional information |
|---|---|---|---|---|
| Other | DMEM/Glutamax | ThermoFisher | 31966–021 | Section 'EPCAM+ BEC isolation, FACS, and flow cytometry analysis' |
| Other | Collagenase | Merck/Sigma | C9407 | Section 'EPCAM+ BEC isolation, FACS, and flow cytometry analysis' |
| Other | Advanced DMEM/F-12 | GIBCO | 12634010 | Section 'EPCAM+ BEC isolation, FACS, and flow cytometry analysis' |
| Other | Glutamax | ThermoFisher | 35050061 | Section 'EPCAM+ BEC isolation, FACS, and flow cytometry analysis' |
| Other | HEPES | ThermoFisher | 15630–056 | Section 'EPCAM+ BEC isolation, FACS, and flow cytometry analysis' |
| Other | Penicillin/Streptomycin | ThermoFisher | 15140–122 | Section 'EPCAM+ BEC isolation, FACS, and flow cytometry analysis' |
| Other | Dispase II | ThermoFisher | 17105–041 | Section 'EPCAM+ BEC isolation, FACS, and flow cytometry analysis' |
| Other | DNAase | Merck/Sigma | DN25 | Section 'EPCAM+ BEC isolation, FACS, and flow cytometry analysis' |
| Other | *E2f1* ON-TARGETplus siRNAs | Horizon | L-044993-00-0005 | Section 'Silencing of E2F1 in EPCAM+ BECs and organoid formation' |
| Other | Scrambled ON-TARGETplus siRNAs | Horizon | D-001810-10-05 | Section 'Silencing of E2F1 in EPCAM+ BECs and organoid formation' |
| Other | TransIT-X2 | Mirus | MIR6000 | Section 'Silencing of E2F1 in EPCAM+ BECs and organoid formation' |
| Other | Matrigel | Corning | 356231 | Section 'Culture of mouse liver BEC-organoids from single bile duct cells of digested livers' |
| Other | 1 X B27 | Gibco | 17504044 | Section 'Culture of mouse liver BEC-organoids from single bile duct cells of digested livers' |
| Other | TrypLE | GIBCO | 12605028 | Section 'Culture of mouse liver BEC-organoids from single bile duct cells of digested livers' |
| Other | Palmitic acid | Sigma | P0500 | Section 'Culture of mouse liver BEC-organoids from single bile duct cells of digested livers' |
| Other | Oleic acid | Sigma | O1008 | Section 'Culture of mouse liver BEC-organoids from single bile duct cells of digested livers' |

*Appendix 1 Continued on next page*

*Appendix 1 Continued*

| Reagent type (species) or resource | Designation | Source or reference | Identifiers | Additional information |
|---|---|---|---|---|
| Other | DAPI | ThermoFisher | 62248 | Section 'Liver immunohistochemistry (IHC) and immunofluorescence (IF)' |
| Other | Cell Recovery Solution | Corning | 354253 | Section 'BEC-organoid whole-mount immunofluorescence' |
| Other | Glucose | Agilent | 103577–100 | Section 'Bioenergetics with Seahorse extracellular flux analyzer' |
| Other | Pyruvate | Gibco | 11360070 | Section 'Bioenergetics with Seahorse extracellular flux analyzer' |
| Other | Glutamine | Agilent | 103579–100 | Section 'Bioenergetics with Seahorse extracellular flux analyzer' |
| Other | Hoechst | ThermoFisher | 62249 | Section 'Bioenergetics with Seahorse extracellular flux analyzer' |

