## [Editor Report]

This important study reports that a high-fat diet induces biliary epithelial cell proliferation and suggests this may account for the so-called ductular reaction in advanced fatty liver disease. Convincing data support the finding that E2f1 is required for BEC proliferation in mice fed with HFD, and organoid models indicate that lipid abundance promotes glycolysis in an E2F-dependent manner.

---

## [Decision Letter]

**Decision letter after peer review:**

Thank you for submitting your article "Hepatic lipid overload potentiates biliary epithelial cell activation via E2Fs" for consideration by *eLife*. Your article has been reviewed by 2 peer reviewers, one of whom is a member of our Board of Reviewing Editors, and the evaluation has been overseen by Carlos Isales as the Senior Editor. The reviewers have opted to remain anonymous.

Essential revisions:

1. Epcam+ cells can differentiate into different parts of the biliary tree, it should therefore be clarified which biliary identity the investigated organoids adopt during the differentiation to relate it to the in vivo data.

2. Gene expression profiling, instead of 4 transcripts in Figure 1E-F, of BEC-organoids after FA treatment may uncover broader transcriptional programs. One would expect E2F-driven transcriptional programs to be increased. One would also expect more cholangiocyte-specific markers as opposed to hepatocyte-specific markers for this BEC-organoid model.

3. The authors relate the presented transcriptional and metabolic changes to the initial and early stage of BEC activation associated with NAFLD, however, do not provide a detailed characterization of whether the results shown correspond to the early or late stage of DR. Related to this, while Ncam1 is known to be upregulated in activated BECs, is the level of upregulation similar to in vivo and generally the authors should provide a more complete molecular characterization of the DR signature to support this point.

4. Figure 1H would be strengthened with a quantitative method to assess lipid accumulation within liver-derived BECs, such as performing Epcam+ BEC FACS isolation from HFD-fed mice followed by BODIPY staining and analysis by flow cytometry.

5. The authors state that lipid overload is sufficient to initiate DR without epithelial damage. While indeed no cell death is observed, the authors do not provide any analysis of the consequences of high fatty acid treatment on the biliary epithelium. In Figure 1, the nuclear pattern of FA-treated organoids differs from those in controls. Likewise, the PANCK immunostaining of livers from mice after 15 weeks of HFD seems to indicate that the bile ducts have at least partially lost their epithelial organization. Please clarify possible changes to the epithelium and how this relates to the stage of DR.

6. In Figure 3D-F, similar to in Figure 2C-D, the majority of bile ducts have 0 proliferating cells. A quantitative method as mentioned above would strengthen these findings. Performing EDU staining as done in Figure 2E would also be more convincing.

7. Given that a full knock-out is used to test the E2F1 requirement, it should be addressed whether a non-autonomous role for hepatocytes in BEC activation can be excluded. Moreover, the authors could use this experimental condition to assess the role of the other three E2Fs, which they identified in the transcriptome analysis. Are they upregulated in the HFD condition, if so, what can be concluded about their function?

8. For the experiments shown in Figure 4C, it should be clarified why organoids from HFD BECs are dissociated with the aim to again form organoids in the presence of an FA mix. Would the results be different, or why can the HFD BEC organoids not be treated directly with the FA mix? Also, does this condition alter the organoid forming potential compared to Fig2H?

---

## [Author Response]

Essential revisions:1. Epcam+ cells can differentiate into different parts of the biliary tree, it should therefore be clarified which biliary identity the investigated organoids adopt during the differentiation to relate it to the in vivo data.

In our study, we followed a well-established protocol to culture biliary epithelial cell (BEC)derived organoids that recapitulates BEC activation, in vitro, and allows the clonal and longterm expansion of EPCAM^+^ biliary cells as self-renewing bi-potent progenitors. These cells are capable of differentiating into biliary- or hepatocyte-like cells in vitro*,* when cultured in specific differentiation media, and in vivo, following transplantation (Huch M. *et al.*, Nature, 2013; Huch M. *et al.*, Cell, 2015; Li B. *et al.*, Stem Cell Reports, 2017; Tarlow B.D. *et al.*, Hepatology, 2014; Dorrel C. *et al.*, Genes and Development, 2013; Aloia L. *et al.*, Nature Cell Biology, 2019). In the absence of specific differentiation factors, these BEC organoids remain undifferentiated and proliferate, mimicking the regenerative response coordinated by reactive BECs (Aloia L. *et al.*, Nature Cell Biology, 2019). In this respect, the identity of these undifferentiated BEC-organoids has been systematically characterized by Clever’s laboratory and compared with primary hepatocytes, hepatocyte-organoids (Hu H. *et al.*, Cell, 2018), and differentiated BEC-organoids (Huch M. *et al.*, Nature, 2013). Moreover, Jan Tchorz's laboratory has further demonstrated that these undifferentiated BEC-organoids have a biliary gene signature (Planas-Paz L. *et al.*, Cell Stem Cell, 2019).

As the main objective of our current study was to characterize the biological behavior of progenitor cells in response to chronic fatty acid load, we only used undifferentiated organoids displaying a biliary gene signature and a marked proliferative capacity. We apologize if this was not sufficiently clear in the first version of our manuscript, and we have now further specified this point in the text (page 4, lines 73-77).

Furthermore, to rule out the possibility that the HFD might induce spontaneous differentiation in vivo and in vitro, we have identified several markers from the literature for hepatocytes and BECs (Planas-Paz L. *et al.*, Cell Stem Cell, 2019). We then investigated these genes in our samples, such as EPCAM^+^ cells isolated from CD- and HFD-fed mice livers (Author response image 1) and their organoid cultures (Author response image 1), and confirmed that the diet had no impact on the identity of these cells, which kept their bipotent progenitor phenotype.

**Author response image 1. sa2fig1:** Heatmaps representing the expression of hepatocyte and BEC markers in EPCAM^+^ BECs isolated from livers of C57BL/6J mice fed with CD or HFD for 15 weeks (A) and BEC-organoids derived from the cells described in A (B). Heatmaps represent the Z-score calculated as “-1 * Z-score” from the expression data from RNAseq (A) and the δ CT values from RT-qPCR (B), respectively. *36b4* was used as the housekeeping gene..

2. Gene expression profiling, instead of 4 transcripts in Figure 1E-F, of BEC-organoids after FA treatment may uncover broader transcriptional programs. One would expect E2F-driven transcriptional programs to be increased. One would also expect more cholangiocyte-specific markers as opposed to hepatocyte-specific markers for this BEC-organoid model.

We fully agree with the reviewer that a broad analysis could unveil important information. However, as the scope of Figure 1 was to provide proof of concept that BECs can respond to lipid overload by adapting their cellular metabolism, we only monitored four well-established markers of lipid metabolism known to be activated upon metabolic challenge (Figures 1E and 1F). We opted to perform the more unbiased transcriptomics approach in the in vivo setting, in particular on chronically exposed BECs in HFD-fed mice, to avoid possible confounding factors induced by in vitro growth (e.g., growth factors and glucose-enriched cell culture medium that supports continuous expansion, the intrinsic over-proliferative state of the in vitro organoid culture). As a result, we found E2Fs as a candidate transcription factor family. We strongly believe that this approach is more meaningful as it reflects the in vivo HFD-induced changes in BECs.

3. The authors relate the presented transcriptional and metabolic changes to the initial and early stage of BEC activation associated with NAFLD, however, do not provide a detailed characterization of whether the results shown correspond to the early or late stage of DR. Related to this, while Ncam1 is known to be upregulated in activated BECs, is the level of upregulation similar to in vivo and generally the authors should provide a more complete molecular characterization of the DR signature to support this point.

We thank the reviewers for this relevant comment. To characterize the stage of the DR in our experimental setup, we consulted Dr. Christine Gopfert, an expert pathologist at EPFL, for a blinded analysis of liver sections from CD- and HFD-fed mice. In line with previous HFD studies conducted in C57BL/6J mice (Hebbard L. *et al.*, Nature Reviews Gastroenterology and Hepatology, 2010; Febbraio M.A. *et al.*, Cell Metabolism, 2019; Giles D.A. *et al.*, Nature Medicine, 2017), the histopathological features of the HFD livers were limited to steatosis and inflammation, without any notable detection of fibrosis (Picrosirius red staining) in the portal areas (New Figure 1 —figure supplement 1E, page 6, lines 109-111). In addition, the architecture of bile ducts (pan-cytokeratin (PANCK) and osteopontin (OPN) immunohistochemistry) was similar in CD and HFD livers (New Figure 1 —figure supplement 1I, page 6, lines 111-114), which is consistent with the lack of portal fibrosis and inflammation. These findings strongly suggest that HFD is sufficient to prime the reprogramming of a subset of BECs to a “progenitor” state, a well-known early step in BEC activation and an important prerequisite for installing the DR response later on. Based on this evidence and on the fact that the expression of established DR markers does not change in our RNA sequencing data of EPCAM^+^ cells isolated from CD and HFD-fed mice livers (New Figure 2 —figure supplement 1E, page 7, lines 129-133), we conclude that HFD feeding triggers the plastic conversion of quiescent BECs into proliferative progenitors but is insufficient to support a complete DR response, which requires more parenchymal damage and inflammation. To avoid confusion and to be consistent with the literature, we refer to this event as “BEC activation” and to proliferating BECs as “reactive BECs” (Glaser S.S. *et al.*, Expert Reviews in Molecular Medicine, 2009).

4. Figure 1H would be strengthened with a quantitative method to assess lipid accumulation within liver-derived BECs, such as performing Epcam+ BEC FACS isolation from HFD-fed mice followed by BODIPY staining and analysis by flow cytometry.

We thank the reviewers for this helpful suggestion. We used several approaches described below to clarify and improve this crucial point in our manuscript.

i) We acquired new images from liver sections of C57BL/6J mice fed CD or HFD for 15 weeks using a confocal microscope to show more precisely the localization of BODIPY (lipid staining) signal in the PANCK (bile duct marker) positive cells and replaced the images in the first version of our manuscript with the new ones (New Figure 1H).

ii) We performed a new in vivo experiment to quantify the number of EPCAM^+^ BECs that accumulated lipids (BODIPY^+^) in livers from C57BL/6J mice after 15 weeks of CD or HFD feeding. We designed and tested a new FACS analysis protocol and, using the gating strategy illustrated in New Figure 1 —figure supplement 1L and New Figure 1I, we confirmed that EPCAM^+^ BECs accumulate more lipids after HFD feeding (New Figure 1I, pages 6-7, lines 114-117).

Overall, these new data confirm and significantly strengthen our findings, formally demonstrating, in vivo, that BECs efficiently accumulate lipids upon chronic fat overload.

5. The authors state that lipid overload is sufficient to initiate DR without epithelial damage. While indeed no cell death is observed, the authors do not provide any analysis of the consequences of high fatty acid treatment on the biliary epithelium. In Figure 1, the nuclear pattern of FA-treated organoids differs from those in controls. Likewise, the PANCK immunostaining of livers from mice after 15 weeks of HFD seems to indicate that the bile ducts have at least partially lost their epithelial organization. Please clarify possible changes to the epithelium and how this relates to the stage of DR.

We thank the reviewers for raising these points. To address the comment about the nuclear pattern, we re-analyzed our in vitro findings and acquired supplementary images of the organoids treated with the FA mix. These additional studies clearly demonstrate that there is no change in nuclear morphology (New Figure 1B). The lack of epithelial reorganization was furthermore confirmed in vivo by the pathologist, who did not find any morphological difference in the epithelial structure of the bile ducts of HFD-fed mice (see point 3 above). These observations are fully in line with our previous findings that chronic lipid exposure triggers increased proliferation without necessarily invading the portal plate of the parenchyma. As we noticed that the term “early DR” might raise confusion (see point 3), we decided to rephrase this process as “BEC activation” instead of “early DR”. By using this new description, we hope that we have eliminated any possible confusion linked to the broader definition of the DR.

6. In Figure 3D-F, similar to in Figure 2C-D, the majority of bile ducts have 0 proliferating cells. A quantitative method as mentioned above would strengthen these findings. Performing EDU staining as done in Figure 2E would also be more convincing.

We thank the reviewers for this insightful comment. We would like to point out that the low number of proliferating BECs in the healthy unchallenged liver is in line with the literature (Aloia L. *et al.*, Nature Cell Biology, 2019). On the other hand, we understand the reviewer's concern about the specificity of immunofluorescence quantifications and the request to repeat this experiment in a more quantitative manner. To this end, we optimized a new protocol and quantified proliferative EdU^+^ BEC cells by FACS in a new cohort of C57BL/6J mice fed CD or HFD. These FACS data, now included in the revised manuscript (New Figure 2G, page 8, lines 145-149), confirm our previous immunofluorescence quantifications. Unfortunately, we were not able to repeat this experiment in the *E2f1*^+/+^ and *E2f1*^-/-^ mice, as our collaborators no longer had this strain in their animal facility. Despite this unforeseen event, we hope that the reviewers appreciate the new set of quantifications that further substantiate the pro-proliferative effect of long-term HFD on BECs.

7. Given that a full knock-out is used to test the E2F1 requirement, it should be addressed whether a non-autonomous role for hepatocytes in BEC activation can be excluded. Moreover, the authors could use this experimental condition to assess the role of the other three E2Fs, which they identified in the transcriptome analysis. Are they upregulated in the HFD condition, if so, what can be concluded about their function?

We thank the reviewers for this insightful comment. To rule out a non-autonomous role of hepatocytes in BEC activation, we transiently knocked down E2F1 in EPCAM^+^ BECs sorted from livers of C57BL/6J HFD-fed mice by transfecting E2F1-specific siRNAs (SMARTpool mix, Horizon) (New Figure 3 —figure supplement 1D) and then assessed their effect on organoid expansion. As expected, compared to scrambled siRNA, silencing of E2F1 significantly reduced the proliferation of EPCAM^+^ BECs (New Figure 3 —figure supplement 1E, page 10, lines 186-188), confirming that E2F1 is an essential driver of BEC activation and that this effect is cell-autonomous. We believe this experiment strengthens our initial hypothesis that E2F1 drives BECs proliferation upon HFD feeding. We agree with the reviewers that a systematic dissection of the E2F protein family could further tease out the mechanistic basis. However, these investigations go beyond the scope of the current study and will be the focus of future work in the lab.

8. For the experiments shown in Figure 4C, it should be clarified why organoids from HFD BECs are dissociated with the aim to again form organoids in the presence of an FA mix. Would the results be different, or why can the HFD BEC organoids not be treated directly with the FA mix? Also, does this condition alter the organoid forming potential compared to Fig2H?

We thank the reviewers for allowing us to clarify this point. In Figure 2H, we used organoid forming capacity as a proxy for BEC “progenitor activation” in vivo. To this end, we seeded equal numbers of cells from digested livers of CD- and HFD-fed mice and showed that the increased EdU signal observed after HFD (New Figure 2G) coincides with an increased ability to form organoids (Figures 2H and 2I), indicating that long-term fat exposure in vivo is sufficient to trigger “progenitor activation”. Given the aim of this experiment, we believe that the colony-forming assay of FA-treated expanding progenitors would not be informative. The metabolic investigation (Figure 4C) requires a high number of cells. For this reason, we amplified BEC-organoids after their derivation using the traditional method mentioned above (isolating single cells from digested livers and allowing BECs to expand in vitro as organoids). After expansion, these organoids were digested into single cells, counted, and equal cell numbers were then reseeded into 48-well plates to perform standardized and controlled Seahorse experiments. We hope this information justifies the different protocols used in Figures 2 and 4.